# Cumulus Extracellular Matrix Is an Important Part of Oocyte Microenvironment in Ovarian Follicles: Its Remodeling and Proteolytic Degradation

**DOI:** 10.3390/ijms23010054

**Published:** 2021-12-21

**Authors:** Eva Nagyová, Lucie Němcová, Antonella Camaioni

**Affiliations:** 1Institute of Animal Physiology and Genetics, Czech Academy of Sciences, 27721 Libechov, Czech Republic; nemcova@iapg.cas.cz; 2Department of Biomedicine and Prevention, University of Rome Tor Vergata, Via Montpelier 1, 00133 Rome, Italy; camaioni@med.uniroma2.it

**Keywords:** oocyte–cumulus complex, extracellular matrix, hyaluronan, proteasome

## Abstract

The extracellular matrix (ECM) is an essential structure with biological activities. It has been shown that the ECM influences gene expression via cytoskeletal components and the gene expression is dependent upon cell interactions with molecules and hormones. The development of ovarian follicles is a hormone dependent process. The surge in the luteinizing hormone triggers ovulatory changes in oocyte microenvironment. In this review, we discuss how proteolytic cleavage affects formation of cumulus ECM following hormonal stimulation; in particular, how the specific proteasome inhibitor MG132 affects gonadotropin-induced cytoskeletal structure, the organization of cumulus ECM, steroidogenesis, and nuclear maturation. We found that after the inhibition of proteolytic cleavage, gonadotropin-stimulated oocyte–cumulus complexes (OCCs) were without any signs of cumulus expansion; they remained compact with preserved cytoskeletal F-actin-rich transzonal projections through the oocyte investments. Concomitantly, a significant decrease was detected in progesterone secretion and in the expression of gonadotropin-stimulated cumulus expansion–related transcripts, such as *HAS2* and *TNFAIP6*. In agreement, the covalent binding between hyaluronan and the heavy chains of serum-derived the inter-alpha-trypsin inhibitor, essential for the organization of cumulus ECM, was missing.

## 1. Introduction

More than thirty years ago, Bissell and Barcellos-Hoff [1], in their paper entitled “The influence of ECM on gene expression: Is structure the message?”, proposed that the unit of biological function in higher organisms was larger than the cell itself and that the gene expression was dependent on cell interactions with hormones, substrates, and other cells. In addition, they proposed the “Dynamic Reciprocity Model”, in which the extracellular matrix (ECM) influences gene expression via transmembrane proteins and cytoskeletal components. In turn, the cytoskeletal association with polyribosomes affects mRNA stability and rates of protein synthesis, while its interaction with the nuclear matrix affects mRNA processing and possibly transcription. In addition, Spencer et al. [2] presented a model in which the ECM influenced not only gene expression but also tissue specificity through the action of ECM receptors and cytoskeleton. They suggested a possible transport of ECM signaling molecules into the cell nucleus and concluded that the unit of biological function in higher organisms was the cell plus its microenvironment. The ECM is an important part of oocyte microenvironment in ovarian follicles. In agreement, Wrenzycki et al. [3] suggested that mimicking the in vivo microenvironment, the in vitro microenvironment would improve the quality and developmental potential of the matured oocytes. The ECM is a vital structure with a dynamic and complex organization that can carry out multiple biological activities essential for normal organ development and tissue homeostasis [4]. Furthermore, the ECM remodeling through proteolytic degradation releases growth factors from the ECM reservoir, which can affect epithelial cell proliferation and migration and regulate organ morphogenesis [5]. The purpose of this review is to demonstrate how proteolytic cleavage affects gonadotropin-induced cytoskeletal structure, the organization of cumulus ECM, steroidogenesis, and nuclear maturation. The data provided by this review sheds light on the role of proteolytic cleavage and remodeling of the HA-rich cumulus ECM in ovarian follicles. 

## 2. Hormone-Dependent Oocyte Microenvironment

The maturation of an oocyte inside the ovarian follicle is a hormone-dependent process. The in vivo, gonadotropin-induced resumption of meiosis also occurs under control of specific oocyte factors and/or signals that reach oocytes with precise timing. During the first phase of development, an intensive intercellular communication between the different cell types of the OCC occurs. The cascade of maturation events probably drives the right synchronization between nuclear and cytoplasmic maturation [6]. During the second phase of maturation, cumulus expansion and the organization of ECM becomes a dominant event [7,8]. The findings of Chen et al. [9] propose that the synergistic action of two hormones deriving from the presence of both LH and FSH, is required for optimal expansion of the cumulus in vitro. They have shown that the expansion of mouse OCC in vitro, occurs to a degree comparable to that seen in vivo when OCCs are stimulated with highly purified FSH and highly purified LH, together. The authors have described a model for cumulus expansion in which follicular FSH binds to cumulus cell receptors to stimulate the expression of LH receptors on the plasma membrane of cumulus cells (CCs), rendering them responsive to LH. LH then binds to these novel cumulus cell LH receptors to stimulate the synthesis and secretion of hyaluronan (HA)-enriched cumulus ECM, which in turn is stabilized by serum-derived matrix stabilizing factors. Specifically, their results support the possibility that LH can act directly upon the cumulus cells of isolated OCCs to stimulate optimal synthesis of the HA-rich cumulus ECM, only after FSH-mediated the up-regulation of functional LH receptors [9]. In agreement, Shimada et al. [10] and Prochazka et al. [11] investigated the formation of LH receptors (LHR) in porcine cumulus cells surrounding oocytes. Their results indicated that the addition of FSH to the culture medium produced LHR mRNA expression, which in turn induced an increase in the level of LHR on the cumulus cells. The newly synthesized LHR on the cumulus cells, function as a potent activator of cAMP production. The increase in cAMP in the cumulus cells, results in the production of many potential factors that stimulate cumulus cell expansion, oocyte maturation [12,13], and progesterone production in cumulus cells [10,14]. Vanderhyden and Armstrong [15] indicated the essential role of CCs in promoting the normal cytoplasmic maturation of oocytes for pronuclear formation and subsequent developmental capability. Their results showed that a high percentage (45.6%) of fertilized oocytes that matured in the absence of CCs, presented evidence of abnormal fertilization. In contrast, oocytes that matured in intact cumuli were able to develop to viable fetuses (57.8%) after in vitro fertilization in a proportion similar to ovulated oocytes (55.0%). Importantly, secretion(s) from CCs and/or the intercellular communication between CCs and the oocyte were required for the formation of a morphologically normal spindles; moreover, the time when the CCs were removed from OCCs during in vitro maturation significantly affected the apoptotic response of the oocytes [16]. In agreement, Chen et al. [17] demonstrated that the transition of a subset of maternal mRNAs critical for embryonal development is under the control of somatic cell inputs. They concluded that the ovarian follicular microenvironment and maternal signals, primarily mediated through the granulosa and cumulus cells, were responsible for supporting oocyte growth, development, and the gradual acquisition of developmental competence. Finally, Wrenzycki et al. [3] showed that it is possible to achieve blastocyst rates of up 70 % when in vivo matured oocytes are used. In contrast, if oocytes are matured in vitro, blastocyst rates are only half of those matured in vivo. Thus, embryos generated in vitro still differ from the in vivo counterparts. These authors indicated that maturation microenvironment had a great impact on subsequent developmental competence. In vivo, ECM is a significant contributor to the microenvironment by providing the signals to support differentiated cell functions. 

## 3. Transesterification Process between Hyaluronan and the Inter-Alpha-Trypsin Inhibitor

It has been shown that compact OCCs isolated from ovarian follicles, undergo expansion in vitro when high levels of HA synthesized by cumulus cells are organized into the ECM, both in mice and in pigs [18,19]. Hyaluronan is a glycosaminoglycan, normally synthesized by hyaluronan synthase 2 (HAS2) at the plasma membrane using cytosolic UDP-GlcUA and UDP-GlcNAc substrates and extruding the elongated chain into the extracellular space [20]. 

An important factor involved in the appropriate organization of the ECM, is the serum-derived inter-alpha-trypsin inhibitor (IαI). This molecule is synthesized in hepatocytes and it is formed by one or two heavy chains (HCs) linked to the light chain, the chondroitin sulphate proteoglycan bikunin, and it leaves the liver trough the blood stream. Upon a proper stimulus, IαI is recruited to extravascular sites, as it happens in ovarian follicles, in which the HCs are transferred to the locally synthesized HA to form the HC–HA complexes trough a transesterification process. These complexes aggregate into a “cable-like structure”, which is involved in the organization of the oocyte–cumulus ECM [21]. We and others, investigated both LH signaling cascades and the LH-induced expression of ECM-related components [8,22] that are involved in the organization of the expanded oocyte–cumulus complex [23]. To confirm the formation of these HCs–HA Structures, we measured the level of HA produced by expanded OCC during gonadotropin–stimulated oocyte maturation [19] and detected the IαI family proteins by Western blot analysis in mammalian follicles [24]. We described the proper ECM structure that depends on the covalent binding of HCs of IαI molecules to HA, as the principal component of the expanded cumulus ECM in mammalian OCCs [8,25]. We detected ECM-related transcripts expressed by porcine and mice cumulus cells, such as *HAS2* [26], tumor necrosis factor-inducible gene 6 protein (*TNFAIP6)* [27,28], pentraxin 3 (*PTX3)* [29], and versican (*VCAN)* [30]. In particular, we investigated the spatiotemporal localization of serum-derived IαI components in gonadotropin-stimulated porcine OCCs, by confocal and fluorescence microscopy (see Figure 1 and Figure 2). Here, our results demonstrate in vitro gonadotropin-stimulated OCCs cultured in a serum-supplemented medium that accumulated HA and IαI in the expanded cumulus ECM. In contrast, OCC cultured in a serum-free medium (a polyvinylpyrolidone (PVP)-supplemented medium) were not able to organize HA into cumulus ECM. These results demonstrate that oocyte–cumulus ECM is not formed in serum-free culture conditions, while it is correctly organized in the presence of serum-derived IαI [24,25,31]. In agreement, Vanderhyden et al. [15] described the beneficial effects of serum during maturation since it enhanced the coupling of the CCs to the oocytes and improved the transportation of nutrients, hormones, or factors involved in controlling the rate of maturation. To support this, it was shown that the presence of serum instead of bovine serum albumin, improved the viability of CCs and the completion of the first meiotic division in bovine and hamster OCCs [32]. It indicated that a maturation microenvironment has a great impact on the developmental competence of oocytes, and the cumulus ECM is a significant contributor to this microenvironment by providing specific signals to support different cell functions. In mice, in particular, the complete cumulus expansion and maximum retention of HA in the cumulus ECM occurs when 1% of fetal bovine serum is continuously present during the first 18 h of culture. Irrespective of the culture time, when serum is absent, synthesized HA is primarily released into the culture medium, whereas in the presence of serum, HA is primarily retained in the cumulus ECM ([18]. Thus, these findings also support the hypothesis that the serum factor, identified as the IαI serum protein [24,31], is a structural component of the cumulus ECM (see also Figure 1 and Figure 2). Surprisingly, the addition of exogenous HA or HA oligomers effectively displaces endogenously synthesized HA from the ECM into the medium, thereby preventing OCC from expanding and organize a functional cumulus ECM [18].

## 4. Remodeling and Proteolytic Degradation of the ECM 

The presence of a communication pathway from the ECM to the cytoskeleton, and from there to the nucleus, appears to play a fundamental role in tissue development, differentiation, homeostasis, and in disease progression [2,33]. Any mechanical perturbations taking place in the ECM can be transferred to the cell interior and change the cytoskeleton structure and the activity of proteins. The ECM provides mechanical support and biochemical signals; both of which can affect cytoskeletal structure, chromatin organization, and gene transcription. Cells, in turn, organize and remodel the ECM and thus play an active role in microenvironment and in the formation of their own phenotypes. In ovarian follicles, after gonadotropin stimulus, CCs expand and organize viscoelastic cumulus ECM as specific microenvironment around the oocyte. LH induces ovulatory changes involving synthesis of cumulus expansion-related transcripts, such as *HAS2, TNFAIP6, VCAN,* and *PTX3* [8,30]. Essential HCs from IαI, bind covalently to HA to form the expanded HA-rich cumulus ECM in mice and pigs [24,31,34]. Thus, the serum derived proteins of the IαI family markedly contribute to the microenvironment in which ovulation takes place, and the expanded OCC facilitates the ovulation and subsequent fertilization in the ampulla of the oviduct [35]. Nevertheless, the LH surge stimulates in the preovulatory follicles a cascade of proteolytic enzymes, including a plasminogen activator (PA), plasmin, and matrix metalloproteinases (MMPs). These enzymes bring about the degradation of the perifollicular matrix and, most notably, of the meshwork of collagen fibers, which provides the strength to the follicular wall. The pharmacological blockage of any of these enzymes resulted in the reduction of the ovulation rate [36]. Gjorevski et al. [33] proposed that the structure and biochemical properties of existing ECM components (such as proteins, proteoglycans, and HA) are altered via proteolytic cleavage. To reveal the unknown mechanisms that differentially and temporally regulate ECM organization, we [25,37] and others [38] investigated ECM composition through intrinsic regulatory mechanisms, such as the ubiquitin–proteasome system (UPS). Ubiquitin is a small chaperone protein that forms covalently linked isopeptide chains on protein substrates to mark them for degradation by the 26S proteasome. The 26S proteasome is a multicatalytic protease complex that specifically recognizes and hydrolyzes proteins tagged with ubiquitin chains. The subunits of the 26S proteasome, comprise approximately 1% of the total proteome in mammalian cells; the ubiquitin–proteasome pathway serves as the main substrate-specific cellular protein degradation pathway [39,40]. Proteasome is a major cellular protease complex that controls the concentration and turnover of molecules in the ECM, including certain types of proteoglycans (PG), MMPs, and collagens [38]. To investigate whether this proteolytic pathway is involved in ECM organization, we used MG132, a specific proteasomal inhibitor [41,42]. We demonstrated that the inhibition of proteolytic cleavage during oocyte maturation affected the resumption of meiosis in the oocyte, the expression of cumulus-related transcripts, organization of ECM, cytoskeleton structure, and steroidogenesis [25,37]. We evaluated the meiotic progression in hormone-stimulated porcine OCC cultured with or without MG132 inhibitor. While both the resumption of meiosis in the oocyte and the cumulus expansion were accompanied by the disappearance of the actin microfilament-rich transzonal projections (TZPs; [37,43]), treatment with 10 µM MG132 arrested 28.4% of oocytes in the germinal vesicle stage (GV stage, vs. 1.3% in control). The proportion of GV stage oocytes increased progressively to >90% with an increased concentration of MG132 (20–200 µM). We also evaluated cytoskeletal dynamics. The maintenance of TZPs supports an oocyte meiotic block in porcine OCCs [43]. This inhibitor blocked the extrusion of the first polar body and the degradation of F-actin-rich TZPs interconnecting cumulus cells with the oocyte as well as cumulus expansion in pigs [37]. To find whether proteolytic activity was involved in the action of LH on the resumption of meiosis in rats, Tsafriri et al. [44] used a broad-spectrum metalloprotease inhibitor, GM6001. In agreement, this inhibitor of proteolytic cleavage also prevented the LH-induced resumption of meiosis. Several authors demonstrated the involvement of the 26S proteasome in the regulation of oocyte meiosis in mammals; specifically, in rats, mice and pigs [37,45,46,47,48]. In addition, it has been shown that the ubiquitin–proteasome pathway modulated mouse oocyte meiotic maturation and fertilization via the regulation of the MAPK cascade and cyclin B1 degradation [46]. Proteasome is a major cellular protease complex that controls the concentration and turnover of ECM molecules, and proteasome activity is regulated by PG-derived glycosaminoglycan [38]. Therefore, we investigated the relationship between proteolytic cleavage and the formation of cumulus ECM in porcine ovarian follicles [25]. The formation of expanded HA-rich cumulus ECM depends on HA association with specific hyaluronan-binding proteins [49], such as IαI [24,31], TNFAIP6 [27,28,50,51,52], PTX3 [29,53], and versican [30,54]. While the mRNA expression of *HAS2* and *TNFAIP6* in the gonadotropin-stimulated OCC was increased in pigs [28], mice [50,55], and rats [52], in the presence of the proteasomal inhibitor, the expression of ECM components, such as *HAS2* and *TNFAIP6*, was markedly reduced and no signal of HA was detected in porcine OCCs [25]. 

## 5. Proteolytic Cleavage of the ECM Component Versican 

It is known that LH surge stimulates, in the preovulatory follicles, a cascade of proteolytic enzymes, including PA, plasmin, and MMPs [36]. A disintegrin and metalloprotease with thrombospondin type I motifs (ADAMTS) proteases are secreted MMPs that play key roles in the formation, homeostasis, and remodeling of the ECM. The substrate spectrum of ADAMTS proteases can range from individual ECM proteins to entire families of ECM proteins, such as the hyalectans. ADAMTS-mediated substrate cleavage is required for the formation of an ECM to the needs of individual tissues and organ systems [56]. Interestingly, Russell et al. [54] investigated the localization of ADAMTS-1 and the proteolytic cleavage of the ECM component versican, during cumulus expansion and ovulation. In ovulating OCCs, versican was cleaved and a 70 kDa N-terminal fragment was detected by the DPEAAE specific versican antibody. In agreement, the proteolytic cleavage of versican was demonstrated in the ECM of porcine in vivo, and in vitro expanded OCCs [30]. The proteoglycan versican is another HA-binding component which can affect the stability of the oocyte–cumulus ECM [54,57,58]. This PG has a complex structure constituted by a high affinity HA-binding N-terminal globular domain (G1), a chondroitin sulfate substituted midsection (αGAG and βGAG), and a cell surface- and matrix-interacting C-terminal globular domain (G3) [59]. Alternative splicing events of a single gene [60] generate either a full-length protein possessing both αGAG and βGAG domains (V0~370 kDa), or shorter proteins with the αGAG alone (V2~180 kDa), or the βGAG alone (V1~263 kDa), or neither GAG domains (V3~74 kDa). Interestingly, these forms of splicing produce different effects in the cell microenvironment [61,62,63,64]. In addition, versican is a preferred substrate of specific proteases, such as ADAMTS [65]. A subset of them, including ADAMTS1, ADAMTS4, and ADAMTS5, cleave versican V0 and V1 in the βGAG domain, generating HA-binding fragments, ending with the neoepitope DPEAAE at the C-terminus. The degradation product of V0 has a large size, consisting of a fragment of ~220 kDa carrying several GAG side chains linked to the αGAG domain, while cleavage of V1 generates a small fragment of ~70 kDa consisting mainly of the HA-binding region (G1) of the core protein without GAG side chains attached [65]. The versican G1-DPEAAE, also called versikine [66], is a biologically active molecule in the cell environment affecting apoptosis and angiogenesis during development and tissue remodeling [65,67]. Moreover, its role in the organization of a provisional HA matrix is supported by the evidence that mice null for versican, as well as for HA synthase 2, presented an identical embryonic lethality due to a defect in the development of the heart matrix [68]. 

We found that the porcine cumulus ECM contains two types of DPEAAE-positive G1 fragments derived from versican 1 (VG1): one is specific to cumulus cells while the other is in common with granulosa cells. We detected the DPEAAE-positive ~70 kDa fragment of VG1, generated by ADAMTS cleavage in in vivo expanded porcine OCCs. We confirmed gene expression and protein cleavage in vitro, when porcine OCCs were cultured with gonadotropins in the presence of serum or follicular fluid, which are necessary as a source of IαI. This molecule, provides the HCs that are covalently linked to the HA during the organization of cumulus ECM. Under these in vitro culture conditions, porcine OCCs up-regulated versican V1 mRNA and generated a ~70 kDa VG1 fragment that was strongly evident at 26 h and 44 h of culture. In fact, Murasawa et al. [67] identified a macromolecular complex formed by VG1 aggregates associated to HCs (of IαI), translocated onto HA as provisional matrices formed in inflamed skin. No versican cleavage products were detected in the porcine OCCs stimulated in vitro in the absence of serum or follicular fluid, i.e., without a source of HCs of IαI [30]. As a second cleavage product, we found a lower molecular weight VG1 fragment (~65 kDa) in the OCC matrix extract. This lower molecular weight fragment was likely the result of further degradative processes taking place at the time of ovulation, when great synthesis and the activation of several hydrolytic enzymes occurred in granulosa and cumulus cells [69], leading to the detachment of OCCs from the wall and follicle rupture. 

## 6. Steroidogenesis and the Activity of Proteolytic Enzymes

There is evidence that ADAMTS1 is regulated by progesterone and the luteinizing hormone during ovulation, where it can play a role in versican cleavage with consequence ECM remodeling around the OCCs [70]. The analysis of whole rodent ovaries showed upregulated expression levels of versican V0/V1 and their degradation enzymes, ADAMTS1 and ADAMTS4, after an LH ovulatory surge [54,71]. As reported by Shimada et al. [72], the time course of versican (V1) cleavage correlated with the expression and activation of ADAMTS1. 

Notably, female progesterone receptor knock-out (PRKO) mice have an impaired periovulatory induction of *Adamts1* mRNA [73,74]. Consistent with this finding, *Adamts1* knockout mice showed abnormal ovaries and reduced fertility [75]. Finally, Russell et al. [54] indicate that one function of ADAMTS1 in ovulation is to cleave ECM component versican, and suggest that the anovulatory phenotype of PRKO mice is at least partially due to the loss of this function. Together, PRKO as well as *Adamts1* null mice display infertility due primarily to impaired ovulation. Since PRKO mice are important for understanding the role of proteolysis in ovarian follicles, it is necessary to explain that steroids are involved in follicular growth, somatic-cell differentiation, and the acquisition of the developmental competence of mature oocytes [76]. All steroids derive from cholesterol by the action of a complex set of enzymes that includes cytochrome P450, present in both mitochondria and endoplasmic reticulum. Gonads receive cholesterol from low density lipoproteins. The steroidogenesis is determined by cholesterol import into mitochondria by the steroidogenic acute regulatory protein (StAR). The steroidogenic capacity is determined by the amount of P450scc protein present, which is determined by gene transcription [77]. The terminal differentiation of CCs within the ovarian follicle, plays a crucial role in the ability of the oocyte to resume meiosis [78,79]. Gonadotropins induce the expression of progesterone receptors (PR) in CCs and the increase in progesterone secretion by porcine OCCs [14,80]. Progesterone has been shown to enhance the activity of proteolytic enzymes important for the rupture of the follicular wall at ovulation [81]. The transfer of cholesterol across the mitochondrial membranes is promoted by StAR [82]. The involvement of the proteasome in the turnover of StAR has been described [83,84,85], with the subsequent influence on progesterone synthesis [84]. We have shown that, in proteasome inhibitor MG132-treated porcine OCCs, the progesterone levels were significantly reduced [25]. In the gonadotropin-treated immature rat ovary, Iwamasa et al. [81] suggested that progesterone played an indispensable role during the first 4 h of the ovulatory process by regulating proteolytic enzyme activities. Our results [25] showed that the ability of gonadotropin-stimulated porcine cumulus cells to produce progesterone, to a level comparable with control OCCs, was not restored when MG132 was present for 20 h of culture, but it was restored at 50 % of control when the MG132 inhibitor was present for only 3 h. Recently, Ogiwara and Takahashi [86] have shown that matrix metalloproteinase 15 (Mmp15) is among the proteases involved in follicle layer ECM hydrolysis, the only LH-inducible enzyme in the teleost medaka. They demonstrated that the LH-induced expression of the *mmp15* gene is accomplished in two steps. In the first step, the nuclear progestin receptor Pgr is induced by the LH surge, and the resulting Pgr is then complexed with 17α, 20β-dihydroxy-4-pregnen-3-one, the physiological progestin ligand for medaka Pgr [87,88,89], to become an active transcription factor. In the second step, activated Pgr, together with the transcription factor CCAAT/enhancer-binding protein β, contributes to the expression of mmp15 mRNA [90]. They found that, in medaka, the cyclin-dependent protein kinase (CDK) inhibitor, roscovitine, inhibited not only follicle ovulation, but also follicular expression of mmp15 mRNA, implicating a CDK action in the expression of the protease gene in the follicle. Their results suggest that, after phosphorylation, Pgr becomes a functional transcription factor for mmp15 gene expression, and that Cdk9 and cyclin I are involved in the process of Pgr phosphorylation [86]. Progesterone receptors are well-known ubiquitin substrates [91]. Nuclear receptors are common targets of ubiquitination [92]. PR activity is stimulated by the yeast E3 ubiquitin ligase RSP5, and its human homologs hRPF1 [93] and E6-AP [94]. The co-expression of UBCH7 and E6-AP, enhances transcription by PR synergistically, and SRC-1 co-activation of PR requires UBCH7 [95]. Receptor downregulation is a stimulatory switch that accelerates the “on” and “off” cycling of receptors from pre-initiation complexes required for active transcription. The inhibition of proteasome activity prevents receptor degradation and suppresses PR-dependent transcription [96]. Thus, like other transcription factors, PR degradation is closely linked to high activity. Besides targeting the receptors, proteasomal degradation influences multiple other factors critical to transcriptional activity, including the recruitment of RNA polymerase II to receptor-bound promoters. Thus, PRs are subject to post-translational modifications that control their action in response to progesterone [92].

## 7. Discussion

In this paper, we have shown that proteolytic cleavage is involved in the organization of cumulus ECM, and that the ECM is the appropriate microenvironment for somatic cell differentiation and the acquisition of developmental competence by mature oocytes. In fact, the proteolytic activity of the 26S proteasome is required for the meiotic resumption, germinal vesicle breakdown, and cumulus expansion of porcine OCCs matured in vitro. In addition, proteasome inhibition also affects the progesterone secretion and expression of cumulus ECM-related transcripts, suggesting the requirement of ubiquitin-proteasome pathway-regulated protein turnover for the formation of ECM during cumulus expansion in the preovulatory period in pigs. In the present paper, we provide evidence that porcine OCCs are autonomous in producing and cleaving versican 1 (V1) during the process of oocyte maturation. We show two distinct cleavage products of the G1 domain of V1 accumulated in the cumulus ECM. One of them, the cleaved fragment of ~70 kDa VG1, interacted strongly with the HA-rich cumulus ECM. The second cleavage product, the VG1 fragment (~65 kDa), aside from cumulus matrix, was the only species detected in granulosa cells. These results strongly suggest that this VG1 fragment is involved in stabilizing the HA-rich cumulus ECM structure by interacting with HA–HCs (of IαI) complexes. 

Importantly, in each organ the ECM directs the essential morphological organization and physiological function by binding growth factors and interacting with cell surface receptors to elicit signal transduction and regulate gene transcription [97]. During cancer progression, as well as inflammation, and the cumulus expansion shares similarity with the inflammation process, an extensive remodeling of the ECM composition occurs. Under such conditions, aberrant signaling promotes epigenetic gene alterations, and the expression of growth factors and cytokines that trigger the expression of the key regulatory proteins and ECM molecules [98]. Hyaluronan, as a major ECM molecule, was localized to nucleoli and to areas of condensed chromatin in the nuclear periphery in oocytes and cumulus cells [99]. Kan et al. [99] also found HA to be associated with the rough endoplasmic reticulum. Interestingly, it has been suggested by Evanko et al. [100] that HA and HA-binding molecules have some roles in the nucleus, perhaps in the transportation of proteins targeted for ribosomal production and trafficking, or for mRNA processing. Following transcriptomic analysis, Li et al. [101] found the transcript, *Chsy1*, an enzyme essential for synthesis of chondroitin sulfate synthase 1, one of six glycosyltransferases that catalyze the biosynthesis of chondroitin sulfate proteoglycans (CSPGs). CSPGs, such as versican, are involved in the formation of HA-rich cumulus ECM. The presence of the critical enzyme for CSPG biosynthesis in oocytes, suggests a role in oocyte development, possibly in the biosynthesis of component of the intracellular matrix. Nevertheless, the exact mechanisms by which HA is incorporated in the intracellular compartment, or its function at these sites, are currently not completely clarified and require further investigation. Our recent results suggest that the versican fragment (VG1) is involved in formation of the HA-rich cumulus ECM structure [30]. Furthermore, we described the remodeling role of HA-rich ECM and its proteolytic cleavage in oocyte microenvironment. 

Taken together, these data provide evidence for the essential role of proteolytic cleavage and remodeling of the HA-rich cumulus ECM in ovarian follicles, thus supplying the oocyte with a microenvironment suitable for its future development. 

## Figures and Tables

**Figure 1 ijms-23-00054-f001:**
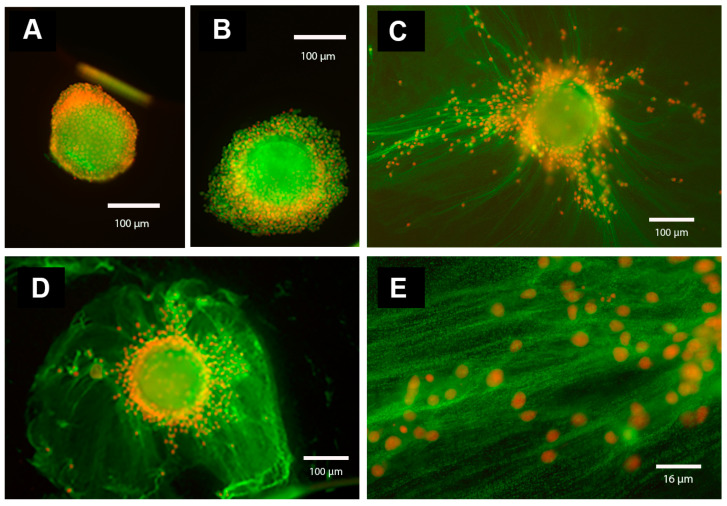
Immunofluorescence analysis of IαI components in the oocyte–cumulus ECM. Oocyte-cumulus complexes (OCCs) were cultured for 20 h without (**A**) or with (**B**–**E**) FSH (100 ng/mL) in the absence (**A**,**B**; in PVP-supplemented medium) or in the presence of serum (5 % FBS) (**C**–**E**, magnification of ECM structure present in **C**). In the absence of serum (**A**,**B**), the cumulus cells remain close to each other and to the oocyte, and the ECM is absent. In contrast, in the presence of serum and FSH (**C**–**E**, magnification of the cumulus ECM structure present in **C**), cumulus cells synthesize and organize an ECM structure that contains components immunoreactive for IαI and present as a network of cables. The immunofluorescence analysis was performed by using a rabbit anti-human IαI antibody (dilution 1:100; DAKO, Carpenteria, CA, USA) and Alexa Fluor 488 goat anti-rabbit IgG as a secondary antibody (dilution 1:500; Molecular Probes, green). Nuclei are in red.

**Figure 2 ijms-23-00054-f002:**
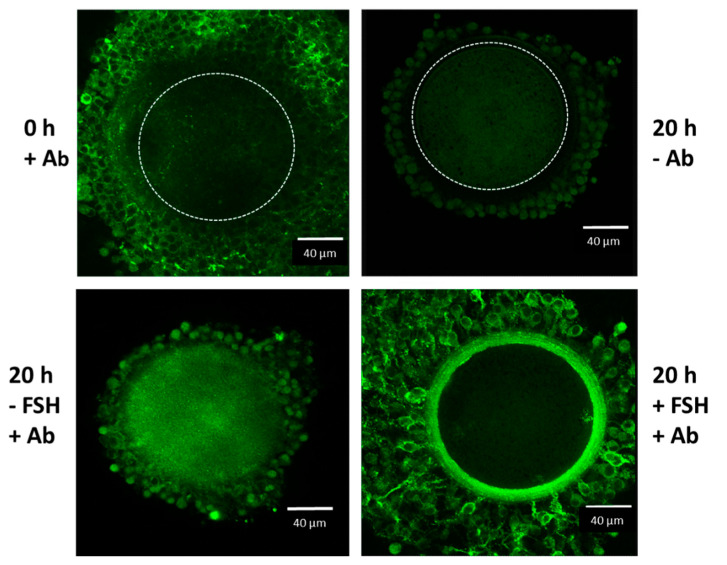
Confocal microscope analysis of IαI in the oocyte-cumulus complexes. Oocyte–cumulus cells complexes were analyzed at the beginning (0 h) and at the end (20 h) of the culture carried out in the presence of 5 % FBS, with or without FSH (100 ng/mL) stimulation, as indicated. Immunoreactivity for IαI components was evident in the ECM structure organized around the cumulus cells after FSH stimulation by using a rabbit anti-human IαI antibody (Ab dilution 1:100; DAKO, Carpenteria, CA, USA) and Alexa Fluor 488 goat anti-rabbit IgG as a secondary antibody (dilution 1:500; Molecular Probes, green). The boundary of the oocyte is marked by a dashed line.

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
