# Peer review of "Cumulus Extracellular Matrix Is an Important Part of Oocyte Microenvironment in Ovarian Follicles: Its Remodeling and Proteolytic Degradation"

_ijms, 2021, doi:10.3390/ijms23010054_

Round 1
Reviewer 1 Report
This is a narrative review where authors tried to describe how proteolytic cleavage affects formation of cumulus ECM following hormonal stimulation. They concluded that proteolytic cleavage is involved in the organization of cumulus ECM, and that the ECM is the appropriate microenvironment for somatic–cell differentiation and the acquisition of developmental competence by mature oocytes. The data provided by authors have shed light to the role of proteolytic cleavage and remodeling of the HA-rich cumulus ECM in ovarian follicles.
There are various points to be reconsidered before reconsideration of this paper as suitable for publication in the journal.
The main point is the review structure. Although this is a narrative review, there is absence of the necessary sections. I would suggest authors to take consultation from relevant papers published in the international literature, as the topic is of interest and resubmit the paper.
Reviewer 2 Report
8th November, 2021
Review of the Manuscript ID: ijms-1462549, by E. Nagyová et al., entitled: “Cumulus extracellular matrix is an important part of oocyte microenvironment in ovarian follicles: its remodeling and proteolytic degradation” that is intended to be published as the Review in International Journal of Molecular Sciences
(separate Microsoft Word file as Reviewer Attachment for Manuscript ID ijms-1462549 IJMS 8th November 2021 that includes Comments to the Authors is also uploaded)
Taking into consideration research highlight, contribution of the Authors to the progress in the research domain, thorough manner of data presentation, relatively well writing in English, and the abundance of manuscript thematic subsections and Figures (diligent graphic visualization), the quality of this paper deserves praise and merits my support. The Authors have received the high scores from me for the originality, importance of the work and the scientific value of their paper. In my opinion, the current paper provides insightful interpretation of topical and coming trends in recognizing molecular mechanisms and factors that determine proteomic rearrangements and proteolytic biodestruction of cumulus oophorus-oocyte complex (COC)-related extracellular matrix within ovarian follicles of different mammalian species. For all these aforementioned reasons, I strongly recommend the Editorial Board to allow for publication of this tremendously valuable paper in International Journal of Molecular Sciences, after the minor revision of the manuscript will have been completed by the Authors and provided that the Authors are ready to consider all the Reviewer comments indicated below:
1) There is a lack of the separate Abbreviations subsection in the paper. That is why, this subsection should have been added at the end of the manuscript to comprehensively elucidate and expand a wide range of the in-text abbreviations, which have been used by the Authors in all the subsections of their paper.
2) The References section has to be prepared in the format compatible with the requirements of International Journal of Molecular Sciences.
General Comment of the Reviewer:
Before the manuscript will have been accepted for publication in International Journal of Molecular Sciences, it requires the minor revision (according to all the remarks and suggestions of the Reviewer).

Author Response
We have shown that the cumulus extracellular matrix (ECM) is an important part of oocyte microenvironment in ovarian follicles. In this review we discuss how proteolytic cleavage affects formation of cumulus ECM, in particular, how the specific proteasome inhibitor affects gonadotropin-induced cytoskeletal structure, organization of cumulus ECM, steroidogenesis, and nuclear maturation.
We greatly appreciate useful comments from the reviewers and their overall positive evaluation of our paper. Therefore, following their suggestions aimed to increase the comprehension and clarity of the manuscript, we have changed its structure and, in the revised version, we have added the following three new paragraphs: Introduction (why review was prepared), Discussion, and Abbreviations.
Concerning references, we were reading again the instructions for authors (IJMS) to check if in the MS all references were correct. And we confirm that they were correct. We have used Zotero, since both Endnote and Zotero bibliography software package is recommended for preparing references to avoid typing mistakes and duplicated references.
We are grateful to reviewers for all their suggestions, for time which they spent to read our paper and for great evaluation.
“The Authors have received the high scores from me for the originality, importance of the work and the scientific value of their paper”. (Rev II)
“The data provided by authors have shed light to the role of proteolytic cleavage and remodeling of the HA-rich cumulus ECM in ovarian follicles” (Rev I)
Thank you.
Reviewer 3 Report
The legends to the figures require rewriting. Currently the reader dose not know what is what , where Thera are OCCs? what is the round structure in the middle? is the a single cell? the legends have to explain what is exactly on a the given panel.
Round 2
Reviewer 1 Report
This is an unstructured narrative review that in terms of methodology and reporting is lacking power.
On the other hand, it is up to the editor decision to accept it, as it contains hard work and valuable information.
The current tendency of scheduling, reporting and especially reading reviews demands different kind of processes.
Reviewer 3 Report
The sentences in the Fig.1 legend. " In the absence of serum (A and B), the cumulus cells remain close to each other and to the oocyte without an ECM structure. On the contrary, in the presence of serum and FSH" has to be changed to: "In the absence of serum (A and B), the cumulus cells remain close to each other and to the oocyte, and the ECM is absent. In contrast, in the presence of serum and FSH "
The sentence in the Fig. 2 legend " In A and B the limit of the oocyte is evidenced by a dashed line" has to be changed to: "In A and B the boundary of the oocyte is marked by a dashed line"
